# The Efficacy of Selected Probiotic Strains and Their Combination to Inhibit the Interaction of Adherent-Invasive *Escherichia coli* (AIEC) with a Co-Culture of Caco-2:HT29-MTX Cells

**DOI:** 10.3390/microorganisms12030502

**Published:** 2024-02-29

**Authors:** Georgia Bradford, Behnoush Asgari, Bronwyn Smit, Eva Hatje, Anna Kuballa, Mohammad Katouli

**Affiliations:** 1School of Science, Technology and Education, and Centre for Bioinnovation, University of the Sunshine Coast, Maroochydore DC, QLD 4558, Australia; georgia.bradford@research.usc.edu.au (G.B.); basgari@usc.edu.au (B.A.); bronwyn.smit@gmail.com (B.S.); 2School of Biomedical Sciences, Faculty of Health, Queensland University of Technology, Brisbane, QLD 4000, Australia; e.hatje@qut.edu.au; 3School of Health and Behavioural Sciences, University of the Sunshine Coast, Maroochydore DC, QLD 4558, Australia; akuballa@usc.edu.au; 4Servatus Biopharmaceuticals, Coolum Beach, QLD 4573, Australia

**Keywords:** AIEC, probiotic combination, Caco-2:HT29-MTX cells, pre- and co-inoculation

## Abstract

The gastrointestinal tract’s microbiota plays a crucial role in human health, with dysbiosis linked to the development of diseases such as inflammatory bowel disease (IBD). Whilst the pathogenic mechanisms underlying IBD remain poorly characterised, adherent-invasive *Escherichia coli* (AIEC) has been implicated as a microbiological factor in disease pathogenesis. These strains show an enhanced ability to diffusely adhere to and invade intestinal epithelial cells, along with the ability to survive and replicate within macrophages. Probiotics, such as *Lactobacillus* strains, have been identified as potential treatment options due to their abilities to compete with pathogens for binding sites and regulate the host immune response. In this study, we used four well-characterised *Lactobacillus* strains and their combination to test their ability to inhibit the adhesion, invasion, and translocation of a well-characterized AIEC strain, F44A-1, in a co-culture of Caco-2 and HT29-MTX cell lines representing the gut epithelium. The results demonstrated that the pre-inoculation of the probiotic candidates 90 min prior to the introduction of the AIEC was more effective in inhibiting AIEC interaction than the co-inoculation of the strains. While the individual probiotic strains greatly reduced AIEC colonisation and invasion of the co-cultured cells, their combination was only more effective in reducing the translocation of the AIEC. These results suggest that probiotics are more effective when used prophylactically against pathogens and that the combination of strains may enhance their efficacy against AIEC translocation once used as a prophylactic measure.

## 1. Introduction

The mucosal layer and microbiota of the gastrointestinal (GI) tract have a substantial impact on human health. When functioning normally, the GI tract is responsible for digestion, absorption, secretion, homeostasis, and acting as a protective barrier against luminal contents and enteric pathogens [1,2]. However, any abnormalities or dysbiosis in the mucosal layer or gut microbiota may lead to the development diseases such as inflammatory bowel disease (IBD), presenting as Crohn’s disease or ulcerative colitis [3,4,5,6]. Whilst the aetiology of IBD is yet to be confirmed, studies have found associations between dysbiosis in the GI tract or a decrease in beneficial bacteria and lower diversity of the microbiota with the pathogenesis of these disorders [7,8,9].

*Escherichia coli* strains, although constituting less than 1% of the gut microbiota, have been found in the GI tract and faeces of patients suffering from forms of IBD, with studies suggesting it has a contributing role in the aetiology of the disease [3,10,11]. Over the past two decades, a pathotype of *E. coli* referred to as adherent-invasive *E. coli* (AIEC) has been frequently isolated from the mucosal membrane of patients with Crohn’s disease [12,13]. These strains can adhere to and invade mucosal membranes and survive and multiply within macrophages [13,14,15]. AIEC causes further dysbiosis through competitive exclusion of both commensal and beneficial bacteria, as well as expressing certain virulence genes that aid its adherence and invasion capabilities [3,8,14,16]. Once AIEC adheres to the mucosal membrane, it can trigger the release of tumour necrosis factor (TNF) and other pro-inflammatory cytokines, leading to inflammation and enhanced epithelial permeability, allowing for further colonisation by the pathogen [7,8,10].

Currently, there are no curative treatments for these disorders, only management options that have small reductive impacts on patients’ symptoms [17,18,19]. The significance of the lack of treatment options, combined with problems arising with increased antibiotic-resistant bacteria has led to research into the use of beneficial bacteria [17,20,21,22]. Probiotics have been described by the World Health Organisation (WHO) as “live organisms that when administered in adequate amounts, can confer health benefits onto the host” [23]. Probiotic strains belonging to *Lactobacillus*, *Bifidobacterium,* and *Propionibacterium* have been found in the GI tract, fermented foods, and dairy products, and over the years have been commercialised to be consumed via tablets or capsules [21,24,25,26]. Studies into both approaches uncovering health benefits include alleviation of constipation and diarrhoea symptoms, improving lactose absorption and antibiotic efficacy, as well as enhancing immune responses and reducing bacterial infections [27,28,29,30].

Probiotics have also been shown to compete with pathogenic bacteria for binding sites and nutrients, leading to adherence to the epithelial or mucosal layers [31,32,33]. Competitive exclusion and adhesion are two of the most effective methods used to block pathogen colonisation, with an added benefit of enhancement of the epithelial barrier [34,35,36,37]. Secretion of antimicrobial substances is another major mechanism, with bacterial fermentation leading to the production of antimicrobial substances. Organic acids, hydrogen peroxide, diacetyl, reuterin, and bacteriocins are some of the substances that aid in inhibition and production of an antagonistic environment against pathogens [20,38,39,40]. Although probiotic strains have a variety of mechanisms to inhibit colonisation of pathogens, these abilities are strain-specific.

There is a plethora of studies that have already characterised the efficacy of single and mixtures of probiotic strains and the benefits they confer on human health, and their use as preventative or therapeutic options [41,42,43]. However, due to the heterogeneity of probiotics in these studies, there is now a move towards combinations of probiotic strains to determine if they are of a greater advantage [27,44,45]. Early research has shown that when in combination, it is possible that a higher efficacy of pathogen inhibition can occur [26,38,46], and that their greater activity is due to a synergistic effect created by the combination of strains [27,44,45,47,48]. Clinically, a higher inhibition of pathogens creates a wider range of health benefits, with some studies already showing greater probiotic adhesion rates, and is linked to alleviation of gastrointestinal disorder symptoms and increased intestinal regularity [45,49,50], although these results may be strains specific. In this study we aimed to test the efficacy of four previously characterised *Lactobacillus* strains with probiotic activities [51] and their combination to inhibit interaction of an *E. coli* strain F44A-1 with all AIEC characteristics [52], using a co-culture of Caco-2 and HT29-MTX cell lines that closely resemble the GI mucosal membrane.

## 2. Materials and Methods

### 2.1. Probiotic Strains

The four probiotic strains selected for this study were *Lactobacillus brevis* M4-205, *L. reuteri* M4-100, *L. rhamnosus* M4-195, and *L. plantarum* M4-165. The isolates were collected from faecal samples of healthy individuals aged between 2 and 40 years old [51] and identified to the species with 16S rRNA as described before [53].

Stock cultures were maintained at −80 °C in de Man, Rogosa, and Sharpe (MRS) broth (Oxoid, Thebarton, Australia) with 20% glycerol and the working cultures were cultured anaerobically in MRS broth or MRS agar plates for 48 h at 37 °C. To ensure strains were pure before each assay, a single colony was selected and inoculated into fresh MRS broth and grown anaerobically. The strain suspensions were centrifuged at 3500 rpm for 5 min and resuspended in phosphate-buffered saline (PBS; pH = 7.00). The concentration of each probiotic strain was adjusted to an optical density of 1.0 at OD^600^nm (approx. 10^9^ CFU/mL) and diluted before the inoculation of wells to 10^7^ CFU/mL. For the combination, equal concentration of each *Lactobacillus* strains (10^7^ CFU/mL) was added to a tube, thoroughly mixed, and inoculated into the wells in each assay.

### 2.2. E. coli Strains

A wild type *E. coli* strain (F44A-1) initially isolated from a patient with colorectal cancer was used as a challenging strain. This strain was characterised for the presence of all virulence genes involved in the pathogenesis of AIEC strains, specifically their diffuse adhesion pattern to Caco-2 cells, as well as their intramacrophage survival and replication [52,54]. As a negative control for adhesion and invasion assays, we used the K12 *E. coli* strain 46-4 [55], and for translocation experiments, we used a highly translocating *E. coli* strain HMLN-1, as a positive control and a non-translocating *E. coli* strain JM109 [55,56,57]. Stock cultures of the strains were maintained in tryptone soy broth (TSB, Oxoid) with 20% glycerol at −80 °C, before being grown in Luria-Bertoni (LB) broth (Millipore, Burlington, MA, USA) in a reciprocal shaker (138 strokes/min^−1^) at 37 °C for 24 h before each assay. The bacterial suspensions were centrifuged at 3500 rpm for 5 min and resuspended in PBS (pH = 7.00) and the cultures adjusted to an optical density of 1.0 at OD^600^nm (approx. 10^9^ CFU/mL) and diluted before the inoculation of corresponding wells to 10^7^ CFU/mL. The multiplicity of infection (MOI) was calculated at 100 CFU/cell, and this was achieved by dividing the number of bacteria inoculated in each well per number of cells. This MOI was used in all infection experiments described hereafter.

### 2.3. Co-Culture of Cells

A co-culture of two human colon adenocarcinoma cells, Caco-2 (ATCC = HTB-37) and HT29-MTX cells. The HT29-MTX cell line is a mucus producing cell lone which was differentiated from HT-29 cells (ATCC = HTB38) into mature goblet cells using methotrexate (MTX) that produce mucin (HT29-MTX). These cell lines express multiple structural and physiological features of the GI tract epithelial layer, including dense tight junctions (TJ), microvilli, and enzymatic activity. However, Caco-2 cells do not produce mucus, an essential feature of the epithelial and mucosal layers of the GI tract. In contrast, HT29-MTX cells can differentiate into mature goblet cells following methotrexate (MTX) treatment [58,59]. Initially, each cell line was grown individually in Eagles Minimum Essential Media (EMEM, Gibco, Thermo Fisher Scientific, Waltham, MA, USA), with Caco-2 cell media supplemented with 20% foetal bovine serum (FBS, Gibco) and HT29-MTX cell media with 15% FBS for growth, and 1% penicillin-streptomycin for both. The cells were incubated at 37 °C in a 5% CO_2_ environment, with the medium replaced every 48 h. Once confluent, each cell line was dissociated from the culture flask and their concentration adjusted to the 9:1 ratio (Caco-2:HT29-MTX) for inoculation into wells for each assay.

### 2.4. Adhesion Assay

The adhesion of AIEC in the presence and absence of four probiotics and their combination was tested in 8-well chamber slides (Nunc Lab-Tek II, Thermo Fisher Scientific, Waltham, MA, USA) using the method adapted from Smit et al. [54]. Co-culture cells were grown to >70% confluence, washed three times with sterile PBS and the media replaced with 200 μL of antibiotic-free EMEM. In the co-inoculation assay, 100 μL of the AIEC strain and each of the probiotic strains alone and in combination were inoculated into the wells (competitive adhesion assay) at a final concentration of 10^7^ CFU/mL. Chamber slides were then incubated for 90 min at 37°C in a 5% CO_2_ environment. In the pre-inoculation assay, all probiotic strains alone and in combination were initially inoculated into wells and incubated for 90 min at 37°C in a 5% CO_2_ environment. Following initial probiotic incubation, 100 μL aliquots of AIEC were added to each well of the chamber slides containing probiotic strains and incubated for a further 90 min. All wells were washed three times with PBS to remove non-adhering bacteria and cells were fixed with 95% ethanol for 5 min, and Gram-stained to differentiate between *Lactobacillus* (Gram-positive) and AIEC (Gram-negative) strains. Slides were dried and observed under a light microscope. The ability of the AIEC and the probiotic strains to colonise the co-culture was measured by counting the number of cells showing adhesion among 100 randomly selected cells, whilst the number of adhering bacteria per cell was determined by counting the number of bacteria attached to 25 randomly selected cells showing adhesion. All tests were completed in triplicate, and all results are expressed as mean ± SEM.

### 2.5. Invasion Assay

Invasion of the AIEC alone and in the presence of four probiotics and their combination was tested as described previously [60]. The co-cultured cells are grown in a flat-bottom 96-well plate to confluence over 4–5 days. Before inoculating the bacterial strains, the media was replaced with antibiotic-free media and the wells were inoculated either with AIEC alone, or AIEC with probiotic strains. Co-inoculation and pre-inoculation protocols were the same as described for the adhesion assays. Following incubation, the medium was removed, and cells washed three times with PBS and wells inoculated with 200 µL of EMEM containing gentamycin (150 μg/mL) and the plate was incubated for a further 60 min to kill any extracellular bacteria. The medium was then removed, and cells lysed with 0.1% Triton-X-100 for 15 min to release invading bacteria. The lysed suspension was then serially diluted and plated on MacConkey agar No. 3 plates (Oxoid) and incubated overnight and the number of colonies counted. All tests were done in triplicate and the results recorded as mean ± SEM of invading bacteria (CFU) per well.

### 2.6. Translocation Assay

Translocation of the AIEC alone and in the presence of four probiotics and their combination were tested as described previously by Smit et al. [54] with some modifications. The co-culture of Caco-2:HT29-MTX cells was grown on cell inserts (pore size = 8 µm) (Millipore) placed into 12 well plates and incubated under 5% CO_2_ at 37 °C. The co-culture was grown until a fully confluent monolayer was achieved. The cell layer integrity was determined by monitoring the trans-epithelial electrical resistance (TEER) value (Ω) until it began to plateau, between 14–21 days of growth. Cell-inserts and wells are washed three times with PBS and 200 μL of antibiotic-free EMEM was added in the inserts and 600 μL in the outer wells. Like the adhesion and invasion assays, co-inoculation and pre-inoculation testing took take place, with 100 μL of the probiotic candidates added to the inserts 90 min prior to adding 100 μL of AIEC in pre-inoculation, and simultaneously in co-inoculation experiments. After 30 min of incubation, 100 µL of medium in the outer well was serially diluted and inoculated overnight at 37 °C onto MacConkey agar No. 3 plates. At the same time 100 μL of antibiotic-free EMEM was added to the outer well to replace the sampled medium and the wells were incubated again for a further 90 min for a total of 120 min of incubation after which the number of translocated *E. coli* was counted in 100 μL of the medium from the outer well. All tests were done in triplicate, with the plates to be incubated for 24 h and counted for CFU. Results were expressed as mean ± SEM.

### 2.7. Statistical Analysis

Statistical analysis was performed using GraphPad Prism statistical software (Version 9.0.0). Two-way ANOVA analysis was followed by Tukey’s multiple comparisons testing to determine differences in the number of adhering, invading, and translocating bacteria amongst all treatment groups and controls. Relationships between adhesion, invasion, and translocation was evaluated using Pearson’s correlation coefficient. Differences are considered statistically significant if *p* < 0.05.

## 3. Results

In general, both AIEC and probiotic strains individually colonized the co-culture at a similar level with some variations among the probiotic strains that ranged from 50% to 70%. In the presence of probiotics, there was a significant reduction in the percentage of cells colonized when the wells were pre-inoculated with probiotics. For the co-inoculation studies, strain M4-205 and the combination of the four strains had the greater effect; however, the most effective strain was M4-195 when pre-inoculated, resulting in a reduction in AIEC of 14% (Figure 1a). This made M4-195 the most successful candidate overall in reducing AIEC colonisation in both co- and pre-inoculation assays (Table 1). The number of AIEC adhering to individual co-culture cells was also reduced in the presence of all probiotics, in both pre- and co-inoculation assays. Pre-inoculation of the probiotic strains resulted in a significantly greater reduction in the number of AIEC adhering individual co-culture cells compared to co-inoculation of the same strain (Figure 1b). In the co-inoculation studies, the combination of strains was able to competitively exclude the AIEC better than the individual strains, with a 34% reduction (Table 1). In all, the highest inhibition of the AIEC was seen with strains M4-100, M4-205 and M4-195 in pre-inoculation experiments (Figure 1b).

The addition of probiotic strains greatly reduced the invasion of the AIEC strain. Reduction of invading bacteria (per well) varied little between the probiotics, with all inhibiting AIEC invasion in both co- and pre-inoculation studies (Figure 2). Pre-inoculation of the candidates resulted in greater inhibition of AIEC invasion, and overall, strain M4-205 had the best combined efficacy overall, followed by the combination of strains (Table 1).

The translocation of AIEC strains over 120 min was similar to that of the positive control, HMLN-1 strain, which has been shown in our previous studies to be an efficient translocating strain [57]. In all instances, the presence of probiotics reduced the translocation of the AIEC strain; there were no significant differences between the probiotics or their interaction times (Figure 3). Co-inoculation of the AIEC and the candidates showed that *Lactobacillus* strain M4-205 was the most successful in its inhibition of the AIEC, with nearly complete inhibition of AIEC translocation after 30 min and an 88% reduction after 120 min (see Figure 3a,b). However, all candidates demonstrated significant efficacies against the pathogen, including the combination of strains. Pre-inoculation testing was the more successful method, with consistent reductions in AIEC translocation occurring, and the combination demonstrating the highest inhibitions, apart from *Lactobacillus* strain M4-100, with a 95% reduction after 120 min interaction with the AIEC (see Figure 3c,d).

## 4. Discussion

It is generally accepted that probiotics are beneficial bacteria, competing with pathogens for binding sites and resources, essentially inhibiting them from colonising the gut epithelium [17,20]. Previous studies have shown probiotic strains as both a valuable prevention strategy and a therapeutic treatment option, reducing bacterial interaction with the gut epithelium [54,61,62]. Whilst many studies use a single cell line such as Caco-2 or HT-29 to assess the efficacy of probiotic strains in vitro, the lack of mucin production in these cell lines renders them suboptimal as a representation of the cell model of the GI tract when measuring bacterial colonisation [63]. In this study, we used a co-culture of the Caco-2 cells and HT29-MTX cells, as this cell model is a closer representation of the gut epithelium. This allowed for a much better environment for evaluating probiotic abilities, particularly for determining how well they can competitively bind to receptor sites and exclude pathogenic bacteria from colonisation in the presence of mucin [20,64]. The four *Lactobacillus* strains used in this study have previously been shown to meet all selection criteria for potential probiotics in previous studies and are highly adherent to this cell model [51,65]. The AIEC strain F44A-1 used in this study has been shown to possess all virulence characteristics associated with AIEC, including survival and replication within macrophages [52]. Here, we demonstrated that this strain is highly adherent and invasive; and was able to translocate across the co-culture cell lines, making it an effective representative of AIEC pathotype.

In this study, the probiotic strains were able to colonise the co-culture of cells at a similar level to the AIEC strain, although the level of adhesion varied amongst the *Lactobacillus* strains. Individually, the probiotic strains were able to reduce the AIEC adherence to the co-culture cells, demonstrating their competitive capabilities. However, combining the strains did not significantly improve the efficacy of the probiotics. *L. rhamnoses* M4-195 and *L. brevis* M4-205 were the most efficient at reducing AIEC colonisation and AIEC adherence per cell, respectively. This is consistent with findings from a systematic review which found that single strains were, in general, as effective as multi-strain combinations [43]. Furthermore, there appears to be no relationships between the probiotic strains’ ability to colonise the cells nor their adhering numbers per cell with their inhibitory effects on the pathogen. These findings suggest that the competitiveness of probiotic strains against the pathogen may not be entirely dependent on direct competition with binding sites, with some factors such as their secretion of antimicrobial substances or their ability to regulate the immune response [40,53,66,67] being involved in the efficacy of individual strain. Van den Abbeele et al. have previously shown that over a 24 h period *Lactobacillus* strains could not inhibit an AIEC strain’s adhesion to mucus but did affect the growth and survival of the pathogen [68]. This suggests that the reduction in adhesion observed in this study and others [69,70] occurs at the epithelial layer, not in the mucus compartment.

The results of the invasion assay demonstrated that despite the highly invasive capability of the AIEC strain, all the probiotics strains were able to significantly reduce the invasion of the AIEC strain with *L. brevis* M4-205 being the most successful strain followed by the combination of strains. This links well with the findings from the adhesion assays where the *L. brevis* M4-205 was the most successful inhibitor of AIEC adhesion. This strain has also been shown to be very effective in inhibiting adhesion of other enteric pathogens to cell lines representing the intestinal epithelium [53]. However, we have previously shown that there was no correlation between the F44A-1 AIEC adhesion and invasion of the co-culture cell model, and that this also did not correlate with the adhesion capabilities of live biotherapeutics products tested in that study [54]. For translocation assays we used *E. coli* strain HMLN-1, a highly efficient translocating strain as shown before [56,57,71], as a positive control. The *L. brevis* M4-205 was again the most effective strain in reducing translocation of AIEC F44A-1 strain, but only in co-inoculation experiments as opposed to pre-inoculation experiments. Despite this, all probiotics inhibited or dramatically reduced AIEC translocation. Translocation across intestinal epithelial cells is a key component of AIEC pathogenesis [72,73], so we propose that when investigating the inhibitory effects of probiotics their effect on translocation of pathogens as well as their adhesion and invasion should also be examined. 

Nonetheless, the use of co-inoculation and pre-inoculation assays allowed us to assess the ability of probiotics, alone or in combination, for their competitive and prophylactic abilities against the interaction of the AIEC strain with the gut epithelium. Indeed, pre-inoculation of the co-culture with the probiotic strains generally reduced AIEC interaction with the co-culture more than when the AIEC and probiotic strains were co-inoculated. The higher levels of reduction obtained during pre-inoculation assays indicate that probiotics could be better used as a prophylactic measure, suggestive of them having a greater efficacy to interact with the gut epithelium and bind to mucosal layer prior to invasion of pathogens. The effect of pre-inoculation having a greater inhibitory effect compared to co-inoculation has also been reported in other strains [74] and may suggest that probiotic metabolites have or the early interaction with the co-culture cells have a role in the inhibitory action. 

The probiotic strains used in this study had previously demonstrated their heterogeneity and varying efficacy to inhibit pathogens, with each strain expressing differing characteristics [53,75,76,77]. These studies also suggest the use of a combination of strains to fill in the gaps for a more widespread efficacy against pathogenic bacteria [51,65]. Other studies also suggest that combinations of probiotic strains, can have a greater colonisation ability and higher levels of inhibition of pathogens compared to using the same strains individually [35,50,78,79]. In vivo studies have also found that combinations of probiotic strains can alleviate physical symptoms, such as diarrhoea in patients suffering from GI disorders [44]. However, these studies either used single cell lines to represent the gut epithelium when assessing the efficacy of their strains or used laboratory animals to mimic to assess the therapeutic abilities of their probiotic strains [80]. The co-culture cell lines, i.e., Caco-2:HT29-MTX that we used in this study has been postulated to reflect the cellular components of the gut environment more effectively than the monoculture [81], such as Caco-2 cells or HT-29 cells that are void of mucin. This improved in vitro model for testing interaction of pathogens with the gut epithelium provided a robust approach to investigate the impact of probiotics against pathogens such as AIEC F44A-1 as also shown in other studies [54]. 

Using this co-culture of cell lines, we evaluated the ability of four known probiotic strains alone and in combination to work synergistically to compete with invading pathogens for colonisation of the gut epithelium. However, our results suggest that the combination of the strains, although had a better efficacy during the translocation assay, they did not significantly increase the efficacy of the individual probiotic strains to inhibit or reduce interaction of the AIEC strain with the gut epithelium. These data suggest that the combination of probiotics may work differently in different cell lines and/or against different pathogens. We also postulate the differences in our results with those reported for combination of probiotic might also be due to the differences in the ability of probiotics in combination to compete for receptor sites or in production of antimicrobial substances. From a clinical point of view the use of these strains alone or in combination can alleviate the sign and symptoms of the disease in patients with ulcerative colitis. Although at this stage there are no direct clinical applications from this study, further strain characterisation and in vivo experiments are required to assess the clinical application of these probiotics. In this study we combined all four probiotic strains, which may have resulted in probiotic-to-probiotic competition and thereby not resulted in an increased inhibitory action against AIEC. A future direction for this project is to genetically characterise the probiotic strain and formulate combinations based on the knowledge of the presence of different characteristics (e.g., bacteriocins), etc. Despite this, in our study, we found a greater reduction in adhesion, invasion, and translocation of AIEC when the mixture of probiotics was pre-inoculated. These data were consistent in our previous pre-inoculation studies [54,82], suggesting that probiotics as an individual or combined can be more effective when used as a prophylactic measure. Further strain characterisation is needed to determine a proper combination of strains, as different combinations of strains can produce different results [26,83]. 

## 5. Conclusions

In conclusion, we demonstrated that the four probiotic strains effectively reduced the interaction of the AIEC strain in a co-culture of Caco-2:HT29-MTX cells, with this effect more pronounced when the strains were pre-inoculated onto the cells. This suggests that probiotic strains are more effective when they are used prophylactically. This study also found that, there was no significant difference in the reduction in pathogen interaction when the combination of the strains was used, apart from during translocation. Different levels of inhibition were observed amongst the probiotic strains, with strain *L. brevis* M4-205 showing the greatest reductions against the AIEC strain in both the co- and pre-inoculation studies. Further characterisation of these strains can help towards formulating a more powerful combination of the strains to be used as a treatment and prophylactics option.

## Figures and Tables

**Figure 1 microorganisms-12-00502-f001:**
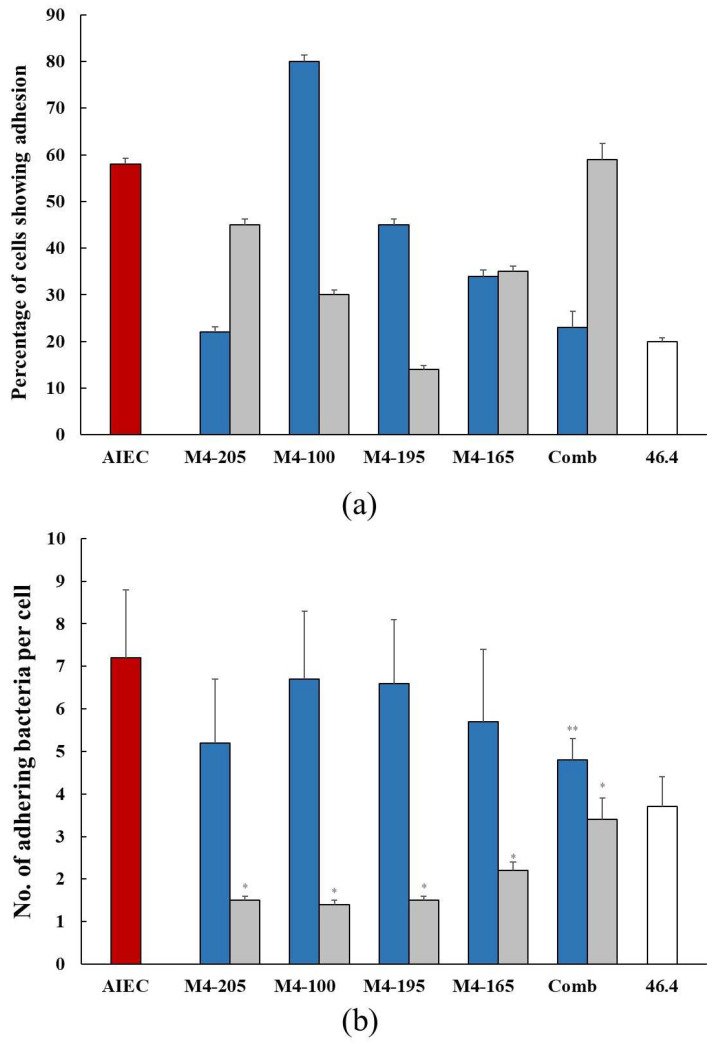
(**a**) The percentage (mean ± SEM) colonization of the co-culture of Caco-2:HT29-MTX cells by the AIEC alone 
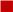
 and in the presence of the *Lactobacillus* strains when co-inoculated 
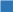
 and pre-inoculated 
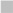
. (**b**) The number (mean ± SEM) of adhering AIEC alone 
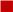
 and in the presence of the *Lactobacillus* strains when co-inoculated 
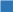
 and pre-inoculated 
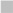
. *E. coli* strain 46.4 used as negative control 
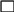
. * *p* < 0.0001 co-inoculation versus pre-inoculation ** *p* < 0.0001 combination versus individual strain.

**Figure 2 microorganisms-12-00502-f002:**
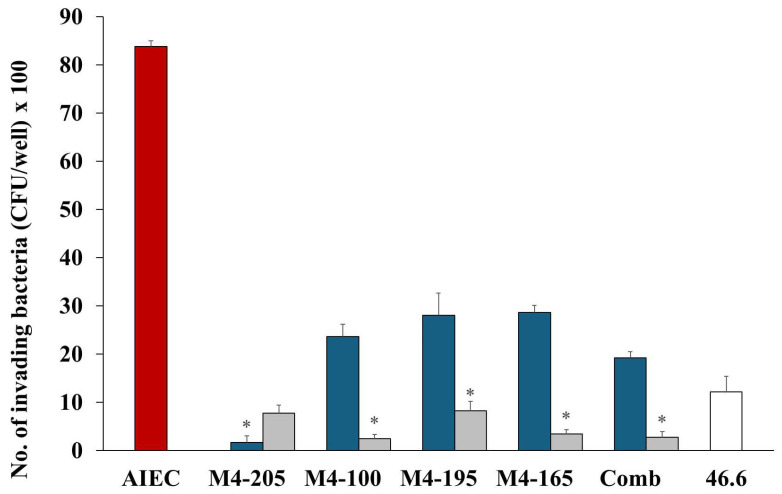
The number (mean ± SEM) of invading AIEC (CFU) alone 
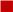
 and in the presence of *Lactobacillus* strains when co-inoculated 
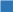
 and pre-inoculated 
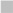
. *E. coli* strain 46.4 used as negative control 
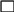
. * *p* < 0.0001 co-inoculation versus pre-inoculation.

**Figure 3 microorganisms-12-00502-f003:**
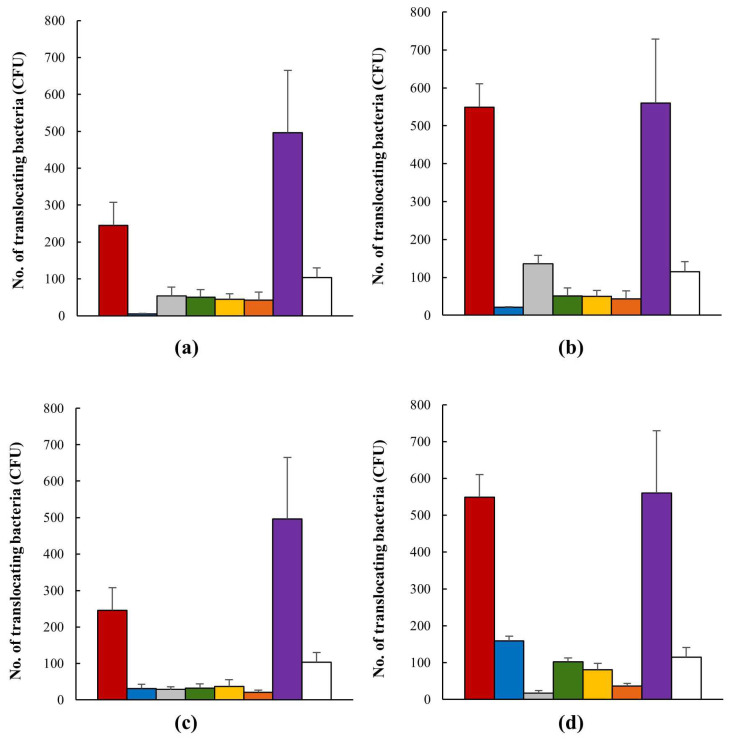
The number (mean ± SEM) of translocating AIEC (CFU) alone 
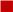
, when in the presence of *Lactobacillus* strains M4-205 
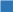
, M4-100 
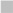
, M4-195 
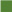
, M4-165 
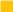
, and their combination 
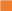
. *E. coli* strain HMLN1 was used a positive control 
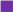
 and *E. coli* strain JM109 as a negative control 
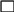
. (**a**) Co-inoculation of AIEC and probiotic candidates after 30 min interaction; (**b**) co-inoculation of AIEC and probiotic candidates after 120 min interaction; (**c**) pre-inoculation of AIEC and probiotic candidates after 30 min interaction; (**d**) pre-inoculation of AIEC and probiotic candidates after 120 min interaction.

**Table 1 microorganisms-12-00502-t001:** Percentage of reduction in colonising AIEC bacteria, and adhesion and invasion of AIEC strain, in the presence of individual probiotic strains and their combination.

Probiotic Candidates	Percentage of Reduction					
	Colonisation		Adhesion		Invasion	
	Co- Inoculation	Pre- Inoculation	Co- Inoculation	Pre- Inoculation	Co- Inoculation	Pre- Inoculation
*Lactobacillus* M4-205	36	13	28	79	98	91
*Lactobacillus* M4-100	-22	28	8	80	72	97
*Lactobacillus* M4-195	13	44	9	80	67	90
*Lactobacillus* M4-165	24	23	21	70	66	96
Combination of four probiotics	35	−1	34	53	77	97

## Data Availability

Data available upon request.

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
