# Peer review of "The Efficacy of Selected Probiotic Strains and Their Combination to Inhibit the Interaction of Adherent-Invasive Escherichia coli (AIEC) with a Co-Culture of Caco-2:HT29-MTX Cells"

_microorganisms, 2024, doi:10.3390/microorganisms12030502_

Round 1
Reviewer 1 Report (Previous Reviewer 1)
Comments and Suggestions for Authors
I am generally satisfied with the comments of the authors and I accept the publication of this work after clarification of 2 points:
The definition of cell differentiation is still unclear. “Differentiation” does not correspond to the treatment of HT29 cells to obtain the MTX clone, but to allow time for the cells to differentiate with an apical pole and a basolateral pole.
Regarding Figure 3, we would have liked to know if the AIEC bacteria do not die partly in contact with probiotics after 30 or 120 min, independently of the eukaryotic cells. This point is not properly addressed.
Author Response
Response to reviewer 1
Comment 1
I am generally satisfied with the comments of the authors and I accept the publication of this work after clarification of 2 points:
The definition of cell differentiation is still unclear. “Differentiation” does not correspond to the treatment of HT29 cells to obtain the MTX clone, but to allow time for the cells to differentiate with an apical pole and a basolateral pole.
Our response
Thank you for your feedback and time taken to read our responses. We are happy to provide this further clarification on your concern regarding cell differentiation. To clarify, the HT29-MTX cells were purchased from a commercial service and are the subclone isolated after MTX treatment that produces mucin – we did not perform the MTX treatment and isolation of the mucin cells. We have shown that mucin is produced, as highlighted in our previous response. With regard to cell polarisation (i.e. the development of the apical and basolateral sides of the cell monolayer), Caco-2 (ATCC= HTB-37) is a polarised cell line. The cells, once grown in EMEM with 20% FBS, produces apical microvilli and tight junction within 2 to 3 weeks and like many of our previous experiments we routinely confirm the production of tight junction (using TEER value readings) and production of microvilli before we use the cells for preparing a co-culture with HT29-MTX at a ratio of 9:1 respectively. There are numerous publications supporting that this process of polarisation that occurs in cell culture of Caco-2 cells.
Comment 2
Regarding Figure 3, we would have liked to know if the AIEC bacteria do not die partly in contact with probiotics after 30 or 120 min, independently of the eukaryotic cells. This point is not properly addressed.
Our response
Thank you for highlighting the aspect of previous comment that you would like further clarification on. It is of course possible that some AIEC strains may die in contact with probiotics after 30min or 120 min. The point however is that for probiotic to effectively produce bacteriocin in a medium such as EMEM, that is not the suitable group medium for their growth (as opposed to MRS medium) the chance that they could produce bacteriocin or any other antimicrobials to kill part of the AIEC is very slim. It is also worth mentioning that the reduction in translocation observed in the presence of probiotics does not necessarily indicate that the AIEC bacteria have died, it is likely that they are viable but unable to establish an initial adherence to the Cacco-2:HT29-MTX cells before invading and to cross the eukaryotic cell monolayer. As part of the future direction of this project we are exploring the direct competition between the AIEC and the probiotic, although this is an experimental condition which does not mimic in vivo conditions.
Reviewer 2 Report (Previous Reviewer 2)
Comments and Suggestions for Authors
The manuscript has already been revised, and the authors made several changes. In general, the manuscript is well written, the structure is adequate. The methodology used is sufficient and up-to-date. The results support the discussion. However, I have the following comments.
I. Minor comments:
1. Improve the writing of the objective of the study.
2. The discussion is good. However, I suggest the authors consider the possible clinical applications.
3. The mechanisms that could explain the effects must be improved, for example: what happens with the inflammatory response?
Author Response
Response to reviewer 2
The manuscript has already been revised, and the authors made several changes. In general, the manuscript is well written, the structure is adequate. The methodology used is sufficient and up-to-date. The results support the discussion. However, I have the following comments.
Minor comments:
- Improve the writing of the objective of the study.
Our response
Thank you for your feedback and time taken to read our responses. This particular comment was also in the previous review. However, we felt that re-wording the aims of the study in the introduction can provide a much clearer description of the aim of the study (see line 86-89).
- The discussion is good. However, I suggest the authors consider the possible clinical applications.
In the discussion we have postulated that the use of these probiotic strains in clinic can alleviate the signs and symptoms associated with ulcerative colitis. However, we feel that at this stage it is too early to make a specific assessment of the clinical application of these probiotics and that further in vivo testing is required. We have incorporated this into the discussion section (see lines 351-354).
- The mechanisms that could explain the effects must be improved, for example: what happens with the inflammatory response?
In a concurrent study we have investigated the regulatory mechanism of our probiotic strains on expression of selected genes involved in the cytokine response and regulation of tight junction genes using RNA-seq. The obtained data are now being processed and the first draft of the manuscript is prepared. For this reason, we would rather not to discuss the anti-inflammatory response of our strains via down regulating the expression of inflammatory markers in this paper.
Round 2
Reviewer 1 Report (Previous Reviewer 1)
Comments and Suggestions for Authors
I'm satisfied with the authors responses.
Author Response
Dear Editor
In response to reviewer's request to improve the quality of the figures and also increase the size of the txt in figures 1 and 2, I am submitting a revised version of the manuscript with improved quality of the figures and also increased sizes of the text which make them easier for the readers to follow the figures.
Thanks
M. Katouli
Corresponding author

This manuscript is a resubmission of an earlier submission. The following is a list of the peer review reports and author responses from that submission.
Round 1
Reviewer 1 Report
Comments and Suggestions for Authors
The paper submitted by Georgia Bradford et al entitled “The efficacy of a cocktail of probiotics to inhibit the interaction 2 of adherent-invasive Escherichia coli (AIEC) with a co-culture 3 of Caco-2:HT29-MTX cells” suggests that probiotics are more effective when used prophylactically against pathogens and that the combination of strains may enhance their efficacy against AIEC translocation once it was used as a prophylactic measure. This work has many weaknesses and the results do not fully support the conclusions. Therefore, this work cannot be accepted in this form.
Major comments:
1. The authors in the introduction talk about the context of IBD, and then use in their work a strain of E. coli isolated from colorectal cancer. Generally, adherent and invasive strains are more associated with IBD, and cyclomodulin-producing strains associated with colorectal cancer. Is the strain used really correspond to the AIEC definition? Furthermore, AIEC bacteria are not characterized by a particular virulence gene, but by the presence of pathoadaptive mutations in different genes, and a particular regulation of the expression of these genes. How can the authors conclude that this strain of E. coli has the AIEC genes? Finally, these experiments must be confirmed with an AIEC strain derived from Crohn's disease patients, otherwise, the conclusions of the paper must be directed towards the pathology of colorectal cancer.
2. The authors use a co-culture model of intestinal cells, but in this model the cells are not differentiated. What is the point of doing this model of co-culture using undifferentiated cells? Do goblet cells have the ability to secrete mucus under these conditions?
3. The authors use a very high MOI in their experiments, MOI = 100. The authors must explain the choice of this MOI because it is not physiological. Under these conditions we can expect all cells to be infected, and therefore minimize the observed effects. The authors need to discuss this point.
4. Concerning the results presented in Figure 1a, no standard deviation is presented, the number of replicates is not specified. It is therefore difficult to interpret the results under these conditions.
5. Concerning the results in Figure 1b, can we consider that the effectiveness observed mainly on the number of bacteria per cell is sufficient for probiotics, knowing that the number of infected cells only drops moderately? For example, the M4-205 strain in pretreatment does not reduce the number of infected cells, so the 1-2 bacteria/cell that remain can multiply and recolonize the cells. The authors should comment on this point.
6. Table 1 does not seem necessary because it simply summarizes the results of Figures 1 and 2. Also explain the meaning of a negative % reduction (-22%). Does this mean you are seeing an increase? How to explain it?
7. Concerning the results presented in Figure 3, several important controls are missing. It is essential to check the number of viable AIEC bacteria after 30 and 120 min of contact with the Lactobacilli strains to know if the bacteria have not died (acidification of the environment by the lactobacilli), which would explain the drop in translocation. Without this control, it is difficult to conclude on the modulation of the translocation capacity of AIECs. In addition, studies of the viability and integrity of eukaryotic cells in the presence of lactobacilli strains alone are lacking.
Minor comments:
- In the introduction, the authors indicate: Organic acids, hydrogen peroxide, diacetyl, reuterin and bacteriocins are some of the substances that aid in inhibition and production of an antagonistic environment against AIEC [20, 38-40]. The references indicated are review but are not specifically related to AIEC.
- In methodology, why are adhesion experiments carried out visually (Gram staining and counting) and not by CFU determination, while invasion is carried out by CFU counting?
Reviewer 2 Report
Comments and Suggestions for Authors
The manuscript is interesting. The methodology used allows us to evaluate the objective of the study. The results support the discussion. However, I have the following comments.
I. Major comments:
1. In the introduction I suggest including a brief paragraph about diet and the intake of probiotics, for example the intake of yogurt or other fermented dairy products.
2. Currently, the Western diet is deficient in prebiotics, therefore it is necessary to correct this problem. I suggest discussing this point.
3. It would be interesting if the authors link the results with the inflammatory response and oxidative stress (include mechanisms)
3. How would it be possible to obtain better results?
4. What nutritional or clinical projections could this study have?
II. Minor comments:
1. Cocktail is an unscientific word. I suggest using another word, for example mixture.
2. Improve the writing of the objective of the study.